# In Vitro Evaluation of the PMN Reaction on a Collagen-Based Purified Reconstituted Bilayer Matrix (PRBM) Using the Autologous Blood Concentrate PRF

**DOI:** 10.3390/biomedicines13051239

**Published:** 2025-05-20

**Authors:** Eva Dohle, Hongyu Zuo, Büşra Bayrak, Anja Heselich, Birgit Schäfer, Robert Sader, Shahram Ghanaati

**Affiliations:** 1FORM, Frankfurt Orofacial Regenerative Medicine, Department for Oral, Cranio-Maxillofacial and Facial Plastic Surgery, Medical Center of the Johann Wolfgang Goethe University, 60590 Frankfurt, Germanybayrak.busra@icloud.com (B.B.); r.sader@em.uni-frankfurt.de (R.S.); s.ghanaati@med.uni-frankfurt.de (S.G.); 2ABIS e.V., Academy for Biological Innovations in Surgery Formally Known as SBCB e.V., Society for Blood Concentrates and Biomaterials e.V., 60435 Frankfurt, Germany; 3Geistlich Pharma AG, 6110 Wolhusen, Switzerland; birgit.schaefer@geistlich.com

**Keywords:** collagen-based Purified Reconstituted Bilayer Matrix, neutrophil priming, cytokines, wound healing, platelet-rich fibrin

## Abstract

**Background/Objectives**: The body’s reaction after the implantation of a biomaterial is a non-specific inflammatory response that is mainly initiated via the recruitment of polymorphonuclear cells (PMNs) to the implant site secreting cytokines and growth factors, followed by activation of monocytes/macrophages, finally leading to wound healing. The wound healing process is dependent on the priming of the PMNs that can be guided towards an inflammatory or a regenerative phenotype with the associated characteristic PMN cytokine profiles. Since the collagen-based Purified Reconstituted Bilayer Matrix (PRBM) triggers the wound healing process at the implant site in vivo, it is hypothesized that this positive effect might be due to a material-mediated priming of the PMNs towards the regenerative phenotype. With the use of the blood concentrate platelet-rich fibrin (PRF) containing high concentrations of leukocytes, including PMNs, the natural environment of the body after the implantation of a material can be mimicked in vitro. The aim of the present study was to characterize the phenotype of native blood-derived PMNs within PRF in response to the PRBM. **Methods**: PMNs within PRF gained from different relative centrifugal forces were characterized in a first step before PRF was combined with the PRBM for 4 h. Supernatants were harvested to analyze the phenotype of the PMNs via the evaluation of eight different cytokines using the ELISA. **Results**: Analysis of the PMN phenotype could assess cytokines commonly associated with neutrophils of the proinflammatory phenotype, such as TNFα, IL15, and IL1, as lower in supernatants when PRF was incubated in the presence of the PRBM and compared to the control PRF. On the other hand, cytokines related to the PMN regenerative phenotype, like TGFβ and IL10, could be detected as higher when PRF was incubated in the presence of the PRBM. **Conclusions**: This might suggest that PRBM significantly activates and primes neutrophils to the regenerative phenotype, leading to the resolution of inflammation. This might trigger the process of wound healing and tissue regeneration, making the PRBM a beneficial material for therapeutic applications.

## 1. Introduction

The initial interaction that takes place when a biomaterial is implanted into the organism is the contact of the material with the surgical wound. The body’s response to this contact is a non-specific inflammatory response. This can range from a mild inflammatory reaction with complete integration of the material to a rejection reaction. Polymorphonuclear cells (PMNs) are the most abundant nucleated cells in circulating blood, characterized by cytoplasmic granules filled with antimicrobial polypeptides. PMNs’ lifecycle involves maturation and release from the bone marrow, followed by circulation in the bloodstream. Upon activation, peripheral PMNs adhere to the vasculature and migrate out of the bloodstream towards sites of infection or inflammation, a process that induces significant changes in their behavior and functional responses [1,2]. After implantation of a biomaterial into the human body, the reaction towards the biomaterial adds to the surgery-related acute wound inflammatory reaction. Hence, PMNs are the first immune cells coming into contact with the implanted biomaterial. The implantation process results in an inflammatory response initiated via the recruitment of polymorphonuclear cells (PMNs) to the implant site, followed by the recruitment and activation of monocytes/macrophages, the latter secreting cytokines and growth factors that induce a wound healing process towards wound closure in the optimal case [3,4,5]. This wound healing process is dependent on the priming of the PMN that can be guided towards an inflammatory or a regenerative phenotype [6]. The identification and understanding of PMN phenotypes and their respective marker profiles depending on their functional states (proinflammatory or regenerative) are therefore essential for distinguishing their roles in orchestrating the process of wound healing. Proinflammatory neutrophils play a key role in the immune response by releasing cytokines and mediators that amplify inflammation and recruit other immune cells. Amongst others, key cytokines produced by the PMNs of the proinflammatory phenotypes typically include TNFα, IL1, IL6, and IL15 [7,8]. On the other hand, regenerative neutrophils are associated with adequate wound healing and tissue repair/regeneration rather than inflammation. Their cytokine profile differs significantly from PMNs of the proinflammatory phenotype as they produce and interact with mediators that promote the resolution of inflammation, tissue remodeling, and regeneration, like IL4, IL10, TGFβ, and VEGF [9,10]. It has been shown that biomaterials can selectively modulate the interaction between the PMNs and their priming at the implant site and can positively influence the wound healing process [11]. Since the collagen-based Purified Reconstituted Bilayer Matrix (PRBM) has been observed to be able to trigger the wound healing process at the implant site [12], it is hypothesized that this positive effect might be due to the material-mediated priming of the PMNs towards the regenerative phenotype. Collagen-based materials have gained significant attention due to their biocompatibility, biodegradability, low immunogenicity, and ability to support cellular adhesion and growth [13,14]. Since collagen is the main structural protein of most hard and soft tissues in animals and in the human body, it is extensively used in tissue engineering, wound healing, drug delivery systems, and regenerative medicine as a scaffold material aiding in the repair and regeneration of damaged tissues [15]. Among natural biopolymers, collagen has also emerged as a leading candidate for electrospinning as collagen-based nanofibers exhibit a high surface-area-to-volume ratio, interconnected porosity, and ECM-like architecture, making them highly suitable for a wide range of biomedical applications [16,17].

In order to be able to make a reliable statement about how the reaction of the body (or the surgical wound) will be upon contact with the biomaterial, the initial interaction between the material and the body (implant bed) can be simulated with the use of platelet-rich fibrin (PRF) in vitro. PRF is an autologous blood concentrate and a complex bioactive system containing the key components for accelerated wound healing, namely, fibrin, platelets, and leucocytes, including PMNs and their associated growth factors [18]. PRF can be obtained by the centrifugation of a patient’s own peripheral blood without additional anticoagulants and its composition can be influenced depending on the relative centrifugation force used [19,20,21]. With the use of PRF, it is therefore possible to mimic the natural environment of the body after the implantation of a material and thus create an in vitro mimic for the initial interaction of the implant bed with the biomaterial. The aim of the present study was to characterize the phenotype of native blood-derived PMNs within PRF in response to the PRBM. Therefore, PMNs within PRF gained from different relative centrifugal forces were characterized in a first step before PRF was combined with the PRBM for 4 h. Supernatants were harvested to analyze the phenotype of the PMNs via the evaluation of eight different cytokines using the ELISA.

## 2. Materials and Methods

### 2.1. Ethics

The preparation and application of PRF in this study were approved by the responsible Ethics Commission of the state of Hessen, Germany (265/17) and all donors gave informed consent to the use of their blood for study purposes.

### 2.2. PRF Preparation

Peripheral whole blood used for PRF production was collected from healthy human donors who agreed to participate in this study. After collecting 10 mL of peripheral blood from the median cubiti vein, the blood was centrifuged at 600 rpm for 8 min (low-RCF PRF) or 2400 rpm for 8 min (high-RCF PRF). To obtain a liquid PRF matrix, plastic-coated tubes (Mectron, Cologne, Germany) were used.

### 2.3. Histological and Immunohistological Staining of PRF Clots

After fixing the coagulated high- and low-RCF PRF in 4% Paraformaldehyde (PFA; Carl Roth, Karlsruhe, Germany), the PRF clots were processed with a tissue processor and prepared for the embedding process with a series of different alcohol concentrations. The PRF samples were then soaked in xylene 3 times for 1 h each and embedded in paraffin. After completion of the paraffin-embedding process, the samples were sectioned with a rotary microtome (SLEE, Nieder-Olm, Germany). Sections were placed in a heating oven (37 °C) for at least 12 h. For each clot, at least 5 slides with a thickness of 2 μm were prepared. One slide of each fixed clot was stained with the haematoxylin–eosin (H&E) staining after deparaffinization with a series of xylene and decreasing concentrations of ethanol. In addition, the different slides were stained immunohistochemically for Neutrophil Elastase (ELA; 1:250; Dako, Jena, Germany). For this purpose, the slides were incubated in 10× Tris EDTA (pH 9) for 20 min in a water bath at 96 °C before the samples were placed under running demineralized water for 10 min. After equilibration in TBS tween wash buffers for 5 min, the sections were incubated with ELA antibody diluted 1:250 in 1% bovine serum albumin (BSA; Sigma-Aldrich, St. Louis, USA) in PBS for 30 min. An amplifier was applied for 10 min, followed by a washing procedure with tris-buffered saline solution. The slides were then processed in a hydrogen peroxide (H_2_O_2_) block for 7 min and washed again in tris-buffered saline (TBS). After adding horseradish peroxidase (HRP) for 10 min, another washing process was performed before the slides were incubated in 3,3′-Diaminobenzidine (DAB) for 10 min. Subsequently, the slides were blued with Mayer’s haematoxylin solution and covered in Aquatex^®^ (Merck, Darmstadt, Germany) to fix the sections with coverslips for visual analysis using a scanning microscope (Eclipse Ni/E fluorescence microscope with a DS-Ri2 camera, Nikon, Düsseldorf, Germany).

### 2.4. Collagen-Based Purified Reconstituted Bilayer Matrix Geistlich Mucograft^®^

The collagen-based Purified Reconstituted Bilayer Matrix (PRBM) Geistlich Mucograft^®^ (Wolhusen, Switzerland) is a collagen matrix that offers an alternative to soft tissue grafts by creating predictable dimensions for soft tissue management. Geistlich Mucograft^®^ is composed of two structures: the compact structure provides stability and allows for open healing; the cancellous structure supports the stabilization of the blood clot and the ingrowth of soft tissue cells. In the present study, the product variant Geistlich Mucograft^®^ Seal with a circular shape of 8 mm in diameter was used.

### 2.5. PRF–PRBM Combination and Cultivation

Liquid low-RCF PRF was produced from at least 6 different donors, as described above. After homogenization of the low RFC PRF, 200 µL of the liquid low-RCF PRF was used to cover one PRBM placed in 24-well plates, followed by 30 min of incubation at 37 °C to allow clotting of the PRF. After clotting, 500 µL of RPMI (Thermo Scientific, Karlsruhe, Germany) per well was added to each PRF-coated matrix. After 4 h of incubation, total supernatants of each experimental group were collected and stored at −20 °C before further processing for determination of the cytokine profile. PRF without the PRBM, as well as PRF incubated in RPMI supplemented with 10 µg/mL Zymosan (Sigma-Aldrich, Darmstadt, Germany) to induce a sterile inflammation [22] and a PRF-covered PRBM incubated with RPMI supplemented with 10 µg/mL Zymosan, served as control groups.

### 2.6. Enzyme-Linked Immunosorbent Assay (ELISA)

Cell culture supernatants were analyzed with ELISA DuoSets^®^ (all purchased from R&D Systems, Minneapolis, MN, USA) to quantify the relative protein concentrations of Neutrophil Elastase (ELA), Interleukin1 (IL1), Interleukin 4 (IL4), Interleukin 6 (IL6), Interleukin 10 (IL10), Interleukin 15 (IL15), Tumor Necrosis Factor alpha (TNFα), Transforming Growth Factor beta 1 (TGFß1), and vascular endothelial growth factor A (VEGF). ELISA DuoSets^®^ were performed according to the manufacturer’s instructions in triplicate per supernatant collected. After overnight incubation of the appropriate capture antibody diluted in PBS at RT, the plates were washed 3 times with 300 μL of wash buffer and blocked with blocking buffer before 100 μL of the samples and the respective standards were added to the plate. After incubation with 100 μL of detection antibody diluted in blocking buffer, the colorimetric reaction generated by the addition of Streptavidin-HRP to the samples was used to visualize the respective cytokine concentrations in the wells. A microplate reader detected and recorded the optical density of each well at 450 nm. Finally, concentrations were determined based on the standard curve values and the results were shown as absolute values in pg/mL and were additionally presented as baseline corrected relative to control (low-RCF PRF).

### 2.7. Determination of the Recovery Capacity for Cytokines in the Presence of the PRBM

For each analyzed cytokine, the PRBM was incubated with 500 µL of 3 different standard concentrations per cytokine for 4 h to determine the recovery capacity of the respective cytokines in the presence of the PRBM alone. After 4 h, it was quantified by standard curve interpolation using GraphPad Prism (version 10.3.1 for macOS, GraphPad Software, Boston, MA, USA) as follows. For each cytokine, a standard curve was generated and a non-linear sigmoidal 4-parameter logistic (4PL) curve fitting was applied to the non-logarithmic concentration data (R^2^ values: IL-15 = 0.9998, VEGF = 0.9970, TGF-β = 0.9998, TNFα = 0.9965, IL-1 = 1.000, IL-4 = 0.9993, IL-6 = 0.9995, IL-10 = 0.9999). The interpolation of measured absorbance values was conducted using the fitted standard curves via automated interpolation to determine the corresponding cytokine concentrations. The percentage of growth factor recovery from the PRBM was subsequently calculated based on the difference between the applied growth factor concentration and the measured release.

### 2.8. Statistical Evaluation

Results were calculated as mean ± standard deviation (SD) and evaluated for significant differences with one-way analysis of variance (ANOVA) using MS Excel (Microsoft Office, Microsoft, Redmond, DC, USA) and GraphPad Prism 9.0 software (Boston, USA); differences were considered statistically significant for * *p*-value < 0.05, ** *p*-value < 0.01, *** *p*-value < 0.001, and **** *p*-value < 0. 0001.

## 3. Results

### 3.1. Characterization of PMNs in High- and Low-RCF PRF

The H&E histological staining of high-RCF PRF was exemplarily chosen to visualize the different cellular and acellular components of the blood concentrate PRF (Figure 1A). In this overview staining, platelets appear as small, dark-staining bodies distributed within the fibrin meshwork. Leukocytes, including lymphocytes, monocytes, and PMNs, can be identified by their nuclei stained blue-purple. Although H&E staining does not provide a detailed differentiation, the distinct nuclear morphology and cytoplasmic features can help identify major leukocyte types exemplarily, as indicated in Figure 1A. Furthermore, red blood cells are evident as round, eosinophilic, pink-stained structures. The fibrin matrix, as a major component of PRF, provides a scaffold for cellular elements and appears as a light pink structure stained by eosin. The PMN marker Neutrophil Elastase (ELA; Figure 1C,D), a serine protease stored in the azurophilic granules of neutrophils, is present in neutrophils at all stages of maturation, in circulating or degranulated PMNs, and in neutrophil-derived extracellular traps (NETs). Immunohistological staining of high- and low-RCF PRF for ELA revealed a higher number of ELA-positive cells in low-RCF PRF (Figure 1C) compared to high-RCF PRF (Figure 1D). Accordingly, the determination of ELA concentration in cell culture supernatants of high- and low-RCF PRF after 4 h of incubation (Figure 1B) confirmed a significantly higher concentration of ELA in supernatants of low-RCF PRF compared to high-RCF PRF. In order to establish the experimental baseline setting with regard to the PRF preparation conditions, the concentrations of different markers for distinctive PMN phenotypes (regenerative and proinflammatory) in supernatants of high- and low-RCF PRF were comparatively analyzed (Figure 2). In general, all analyzed cytokines could be detected and measured in both low- and high-RCF PRF supernatants, except for IL6 in supernatants of high-RCF PRF (Figure 2C). IL15, IL10, and IL4, as well as VEGF, could be found at a similar concentration (not significantly different) in both supernatants (high- and low-RCF PRF), whereas TNFα, IL6, IL1, and TGFß were found at significantly higher concentrations in low-RCF PRF supernatants.

### 3.2. Combination of PRBM with PRF: Effect on PMNs’ Cytokine Release

Since generally higher concentrations of PMNs and their released cytokines could be found in supernatants of pure control low-RCF PRF, the PRBM was combined with low-RCF PRF for 4 h before supernatants were analyzed for cytokines related to the PMNs of the regenerative (Figure 3A–E) and proinflammatory (Figure 3F–I) phenotype. Zymosan treatment of PRF resulted in an activation of PMNs, confirmed by a significantly higher release of ELA, IL10, and TNFα in supernatants of the Zymosan/PRF group compared to the untreated control PRF. In particular, for TNFα and IL-1, an exceptionally high standard deviation was observed for the nine donors initially included in this study, which was mainly due to three donors. The cells of these three donors were found to release low levels of IL-1 prior to activation of the RCF PRF and low ELA, TNFα, and IL-1 after Zymosan activation. Therefore, these three donors were excluded from subsequent analysis and results were analyzed and calculated for six donors only. The full data set, including data from the excluded PRF donors, is provided in Appendix A.

The concentration of TGFß was significantly higher in supernatants when low-RCF PRF was combined with the PRBM compared to the control low-RCF PRF without the PRBM and the Zymosan-treated low-RCF PRF (Figure 3B). Under sterile Zymosan-induced inflammation conditions, the combination of the PRBM with PRF resulted in a higher release of TGFß as opposed to the respective low-RCF PRF, however, this difference was statistically non-significant. Unexpectedly, TGFß concentrations were found to be similar in the control (PRF alone) and positive control (PRF + Zymosan). A similar PRBM-mediated effect could be evaluated for the release of IL10. The highest IL10 concentration could be determined in supernatants of the PRBM/PRF combination and the PRBM/PRF-Zymosan group compared to the other experimental groups (Figure 3C). These differences were highly significant (Figure 3C). The higher IL10 release in Zymosan-treated PRF (without the PRBM) compared to untreated PRF confirmed the Zymosan-mediated activation of the PMNs within PRF. Notably, IL10 concentration was found in the same concentration range in the PRBM/PRF and PRBM/PRF/Zymosan supernatants. For IL4 and VEGF, the additional examined cytokines related to the regenerative phenotype of PMNs, no PRBM-mediated effect could be determined: in both cases, the concentrations did not alter between the different experimental groups (Figure 3D,E).

The analysis of cytokines that are associated with the proinflammatory phenotype of PMNs after combining low-RCF PRF with the PRBM for 4 h revealed a clear pattern for TNFα, IL1, and IL15 (Figure 3F,H,I). After 4 h of incubation of the PRF/PRBM combination, a significant reduction of the concentration of proinflammatory cytokines could be measured in the PRF/PRBM supernatants compared to the supernatants of pure low-RCF PRF and Zymosan-treated low-RCF PRF. The concentration of TNFα was found to be the highest in the low-RCF PRF control group and when low-RCF PRF was treated with the proinflammatory agent Zymosan (Figure 3H). This higher TNFα concentration in these experimental groups was found to be reduced by the combination of low-RCF PRF with the PRBM. The same trend could be observed for IL1. The concentration of the proinflammatory cytokine IL1 was also found to be significantly lower when low-RCF PRF was combined with the PRBM compared to the control low-RCF PRF and to the Zymosan-treated low-RCF PRF. There was no difference in IL1 concentration between control PRF and Zymosan-treated PRF (without the PRBM) (Figure 3I). The IL15 concentration was significantly lower in the PRBM/PRF group compared to the control group with untreated low-RCF PRF alone (Figure 3F). For IL6, no PRBM-mediated effect was determined: the IL6 concentrations did not alter between the experimental groups. (Figure 3G).

The baseline corrected cytokine profile in Figure 4 provides an overview of the PMN cytokine level in the PRF/PRBM combination compared to control low-RCF PRF alone (relative concentration to control = 100%; baseline = 0). The graph highlights the PRBM-mediated upregulation of cytokines related to the regenerative phenotype of PMNs, namely, IL10 and TGFß, and the downregulation of the proinflammatory PMN-derived cytokines IL1 and TNFα in response to combining the PRBM with low-RCF PRF compared to the control low-RCF PRF (zero).

### 3.3. Determination of Cytokine Recovery Capacity in the Presence of PRBM Using the ELISA

To investigate whether the observed downregulation of the cytokines, most prominent for IL1 and TNFα, is due to a potential absorption of the respective cytokines to the PRBM, the cytokine recovery capacity of all analyzed cytokines after 4 h of incubation with the PRBM was examined (Table 1). The results revealed that in the presence of the PRBM, the recovery rate is high for IL1 and IL6 (>90%). The recovery rate of all other tested cytokines under the chosen experimental conditions is low, ranging from 30–50% for TNFα, IL4, and IL10 to <10% for TGF and VEGF.

## 4. Discussion

PMNs are the most abundant immune cells in human blood and the first responders to implanted biomaterials, triggering or resolving the process of inflammation after biomaterial implantation [23]. Upon biomaterial implantation, PMNs rapidly migrate to the sites of implantation, where they orchestrate the early immune response [24]. Their dual role involves initiating inflammation by releasing pro-inflammatory mediators, while also contributing to the resolution of inflammation through the production of anti-inflammatory factors [25]. The nature of their response can be influenced by the properties of the implanted biomaterial. The materials’ surface topography, the structure, the stiffness, and the chemical composition can prime PMNs for either impaired inflammatory activity or timely resolution of inflammation processes [4]. This balance is crucial as an excessive inflammatory response might hinder biomaterial integration while the efficient resolution of inflammation promotes tissue healing and implant acceptance. Understanding the influential role of PMNs in the context of biomaterial integration might offer valuable insights into designing and engineering implants and scaffolds that influence their ability to trigger and resolve inflammation effectively, leading to good material integration and regeneration.

During the present study, the phenotype of native blood-derived PMNs within the human blood concentrate PRF was characterized in response to the collagen-based membrane PRBM. Because the PRBM has been observed to be able to trigger the wound healing process at the implant site [12], this study hypothesized that the positive effect on the wound healing process might be a result of the material-mediated priming of the PMNs towards the regenerative phenotype. To analyze the material-mediated effect on the phenotype of PMNs after implantation in vitro, the initial interaction between the PRBM and the implant bed was simulated with the use of platelet-rich fibrin (PRF). PRF is a second-generation platelet concentrate used in regenerative medicine and dentistry for tissue healing and repair [26]. It is an autologous biomaterial derived from the patient’s own blood, prepared without the addition of anticoagulants or chemical agents [18,27]. PRF has gained attention for its ability to enhance healing through the sustained release of growth factors, its biocompatibility, and its role in promoting cellular proliferation and wound-healing-associated processes, i.e., angiogenesis and osteogenesis [19,28,29]. Due to its key components for accelerated wound healing, namely, fibrin, platelets, and leucocytes, including PMNs and their associated growth factors [18], PRF and its containing PMNs might serve as a model to mimic and analyze in vitro the initial interaction of the implant bed with the biomaterial. According to previous studies, the number of neutrophils within PRF strongly varies depending on the relative centrifugation force used during preparation [30]. Immunohistochemical staining of high- and low-RCF PRF for the PMN marker ELA, as well as determination of neutrophil-released cytokines ELA, TNFα, IL6, IL1, TGFß, IL10, and VEGF concentration in cell culture supernatants of low- and high-RCF PRF, confirmed generally higher numbers of neutrophils and neutrophil-associated cytokines when reducing the RCF for preparation of low-RCF PRF. The higher concentrations of cytokines, growth factors, and proteases like ELA, TNFα, IL6, IL1, TGFβ, IL10, and VEGF in cell culture supernatants of low-RCF PRF compared to high-RCF PRF reflect the functional properties of PRF matrices prepared under different conditions. Although the number of PMNs in low- and high-RCF PRF differs, with a higher amount of PMNs in low-RCF PRF, some of the analyzed cytokines, like IL15 and IL4, are not aligned with the higher cell concentration in low-RCF PRF. In this context, the higher centrifugation force upon PRF preparation might activate the PMNs. In line with this assumption, it has already been shown that neutrophils can be stimulated or activated by physical stress factors, such as centrifugation [31]. The high physical stress caused by centrifugation can lead to changes in the cell membrane and cellular functions, which, in some cases, can trigger neutrophil activation, resulting in the release of granule contents or an increased production of reactive oxygen species [32]. While Neutrophil Elastase (ELA) is a valuable marker of neutrophil activity, it is important to acknowledge that ELA can also be released from degranulated or apoptotic neutrophils [33]. As such, it does not exclusively indicate the presence of viable polymorphonuclear cells (PMNs). To enhance the specificity of neutrophil characterization, the inclusion of additional markers, such as myeloperoxidase (MPO), which reflects intracellular enzymatic activity, or surface proteins like CD66b, which are retained on intact neutrophils, could provide further insights.

Since it could already be shown that the variations in PRF composition influence the biological response in wound healing and tissue regeneration, PRBM-mediated effects were evaluated using low-RCF PRF [20,28,34]. Based on our results, cytokines commonly associated with neutrophils of the proinflammatory phenotype, such as TNF, IL15, and IL1, were found to be lower in supernatants when low-RCF PRF was combined with the PRBM. On the other hand, cytokines related to the PMN regenerative phenotype, like TGFβ and IL10, were shown to be higher in response to a combination of low-RCF PRF with the PRBM. This suggests that the PRBM modifies the response of neutrophils harbored within low-RCF PRF.

Considering the recovery of cytokines when analyzed via the ELISA, the measured decrease in IL1 (recovery rate ca. 95%) concentration within the supernatant may be interpreted as an anti-inflammatory effect of the PRBM on PMNs when present in low-RCF PRF. The reduction of TNFα upon the presence of the PRBM, however, may be a consequence of the binding of TNFα to the PRBM or an interference of the supernatant with the ELISA. Whether or not TNFα (recovery rate ca. 40%) as well as IL15 (recovery rate ca. 30%) are reduced upon PRBM presence needs further investigation, e.g., via immunohistology. Following the same rationale, the measured significant increase in TGFβ (recovery rate 5%) and IL10 (recovery rate ca. 30%) related to the regenerative PMN phenotype may be underestimated and may in fact be much stronger. With regards to IL4 (recovery rate ca. 30%) and VEGF (recovery rate ca. 10%), it could be assumed that both cytokines may also increase upon contact with the PRBM since a decrease in concentration could not be measured but would be expected from the low recovery rate. In summary, data are in line with the hypothesis that indeed the PRBM, when applied together with low-RCF PRF, activates and primes neutrophils to the regenerative phenotype due to a decrease in the release of pro- and an increase in the release of anti-inflammatory cytokines. It is important to consider that the observed changes in cytokine concentrations may be partly attributed to physical sequestration by the PRBM rather than solely to the modulation of cellular secretion. As demonstrated in the cytokine recovery assays, substantial adsorption of several cytokines was observed, indicating that PRBM has the capacity to bind and retain soluble cytokines, potentially leading to an underestimation of their actual concentrations in the supernatant. Consequently, the apparent decrease in pro-inflammatory cytokines, such as TNFα, and the measured increase in anti-inflammatory mediators, like TGFβ and IL-10, may not entirely reflect shifts in PMN secretion behavior. This is particularly relevant for cytokines with low recovery rates, where the true extent of the cellular response may be masked or distorted. Future studies employing complementary techniques, such as immunohistochemistry or the quantification of surface-bound cytokines on the PRBM, will be necessary to distinguish between active modulation and passive adsorption effects. Taken together, these findings underscore the importance of analyzing biomaterial–cytokine interactions when interpreting immunological processes.

Given the fact that collagen devices can serve as a carrier for growth factors [14,35,36], it is tempting to assume that some factors, such as VEGF and TGFβ, after PMN release, may bind to the PRBM collagen. Should this be the case, the PRBM may act as a reservoir for cytokines, particularly those with anti-inflammatory properties. By storing and gradually releasing these signaling molecules upon degradation, PRBM may help to regulate inflammation and promote a more favorable healing environment in surgical or regenerative procedures. Understanding the interaction between PRBM and cytokines could lead to improved therapeutic applications in oral and maxillofacial surgery since controlled cytokine release could enhance tissue integration, reduce excessive immune responses, and support soft tissue regeneration. Collagen-based membranes like the PRBM are widely used in regenerative medicine and tissue engineering, particularly in guided bone and tissue regeneration, since these membranes are biocompatible, bioactive, and capable of modulating the immune response, including the behavior of neutrophils [37,38]. Collagen, as a key structural protein of the extracellular matrix (ECM), has profound effects on neutrophils, influencing their behavior, activation, and role in inflammation and tissue repair [39]. Neutrophils interact with collagen via integrins and other receptors, affecting their activation, adhesion, migration, and functional responses [40,41]. Accordingly, several previous studies documented biomaterial-related properties strongly influencing the neutrophil response, such as the type and origin of the biomaterial, the topography or roughness of the surface, as well as the chemical composition of a material [23,42,43,44]. In this context, materials with low stiffness, hydrophobic materials, and natural materials were able to prime neutrophils to a more anti-inflammatory response and phenotype.

When considering the inflammatory response, PMNs are the first cells that extravasate into injured or inflamed tissue in response to stimuli like pathogen- or damage-associated molecular patterns [45,46]. As a result, they induce the process of inflammation by releasing various inflammatory mediators, recruiting additional PMNs, macrophages, and monocytes, undergoing degranulation, oxidative burst, and NETosis [47]. Nevertheless, the role of neutrophils in the resolution of the inflammatory response is not yet fully understood since scientists have focused more on the proinflammatory functions of neutrophils than on their notably important resolution and regenerative properties. The resolution of inflammation by neutrophils seems to be caused by the release of soluble mediators, as well as apoptotic bodies and other vesicles, modifying the microenvironment to terminate the inflammatory response and leading to tissue homeostasis and regeneration [48]. A general reduction of neutrophil numbers through, e.g., efferocytosis, a process of phagocytosis of apoptotic neutrophils by macrophages, results in the blocking of the release of proinflammatory cytokines. This process causes important anti-inflammatory effects by releasing TGFβ and IL-10, leading to a shift to an anti-inflammatory and regenerative microenvironment, as also observed during the present study [49]. Hence, the clinically observed beneficial effect of PRBM may find its counterfeit in the priming of PMNs towards the regenerative phenotype and provision of an inflammation resolution, as well as a regeneration-triggering environment, from the very beginning of wound healing on. Given that PRF itself has a well-documented beneficial effect on wound healing [5,35,50], we may further speculate that PRBM provides an additional positive influence, potentially enhancing the regenerative effects of PRF.

In conclusion, the hypothesis of this study could be confirmed. Our findings highlight a novel insight since the PRBM not only acts as a physical scaffold but also functions as an immunomodulatory agent by priming innate immune cells at the implantation site, defining the PRBM as a beneficial biomaterial triggering wound healing and regeneration. Furthermore, we suggest that biomaterial modification emerges as an interesting approach to modulate and control the PMN response and might improve the integration of a biomaterial into the tissue. A key novelty of this study lies in demonstrating that modulating PMN behavior via biomaterial design offers a strategic pathway to improve early-phase host responses, which are critical for successful tissue integration. Based on these findings, we propose that biomaterial modification could be an effective tool to direct PMN activation profiles, enhancing the regenerative microenvironment and reducing inflammatory complications. Thus, a classification system to validate biomaterials according to the PMN response would help to develop new biomaterials for different applications. Nevertheless, a potential limitation of the present study is related to the time point of the investigation since only the very early response to the PRBM was evaluated after 4 h of incubation and later time points of the biomaterial-mediated reactions have not been considered. Moreover, it is not clear if the reduction of some cytokines is due to an interference when testing the supernatant in the ELISA or due to absorption to the PRBM collagen. Furthermore, primary cells and especially blood cells can vary significantly depending on the donor due to biological variability. Factors such as age, gender, genetic background, immune status, and overall health condition contribute to differences in cell proliferation, differentiation potential, and cytokine secretion, leading to fluctuations in experimental outcomes in response to external stimuli. Similarly, blood cells, including PMNs, exhibit donor-specific differences in cytokine expression, the inflammatory response, and activation capacity. These variations can result in high standard deviations in experimental data, necessitating careful donor selection and statistical approaches to minimize bias and ensure reproducibility.

## Figures and Tables

**Figure 1 biomedicines-13-01239-f001:**
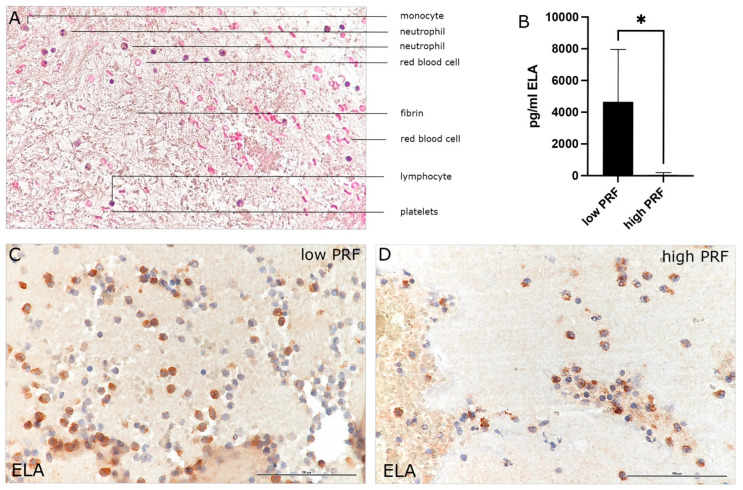
Characterization of PMNs in low- and high-RCF PRF. (**A**) Histological overview staining (H&E) to visualize the cellular and acellular components of high-RCF PRF. (**B**) Determination of Neutrophil Elastase (ELA) in high- and low-RCF PRF. (**C**,**D**) Immunohistological staining for Neutrophil Elastase (ELA) in low- (**C**) and high- (**D**) RCF PRF. Scale bars = 100 µm. * *p*-value < 0.05.

**Figure 2 biomedicines-13-01239-f002:**
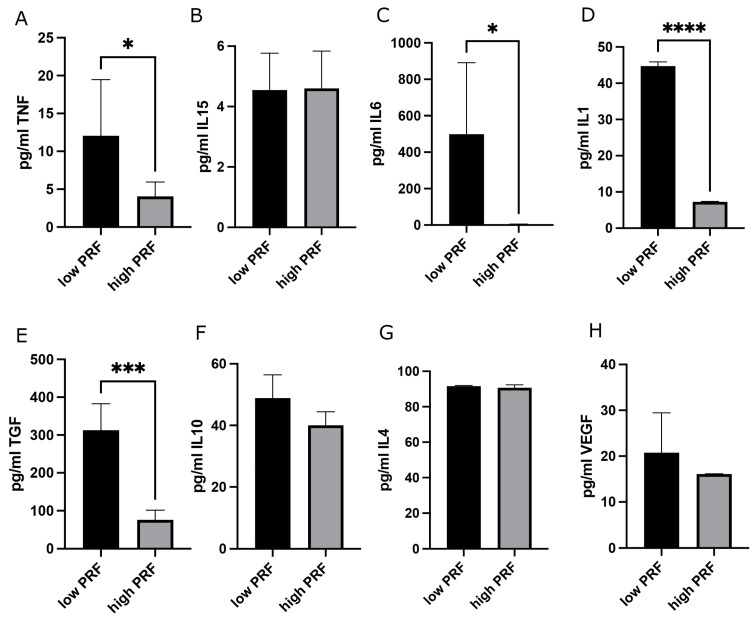
Evaluation of the PMN-related cytokine profile in high- and low-RCF PRF in supernatants via the ELISA. The data show the absolute concentrations of TNFα (**A**), IL15 (**B**), IL6 (**C**), IL1 (**D**), TGFβ (**E**), IL10 (**F**), IL4 (**G**), and VEGF (**H**) in pg/mL. * *p*-value < 0.05, *** *p*-value < 0.001, and **** *p*-value < 0.0001.

**Figure 3 biomedicines-13-01239-f003:**
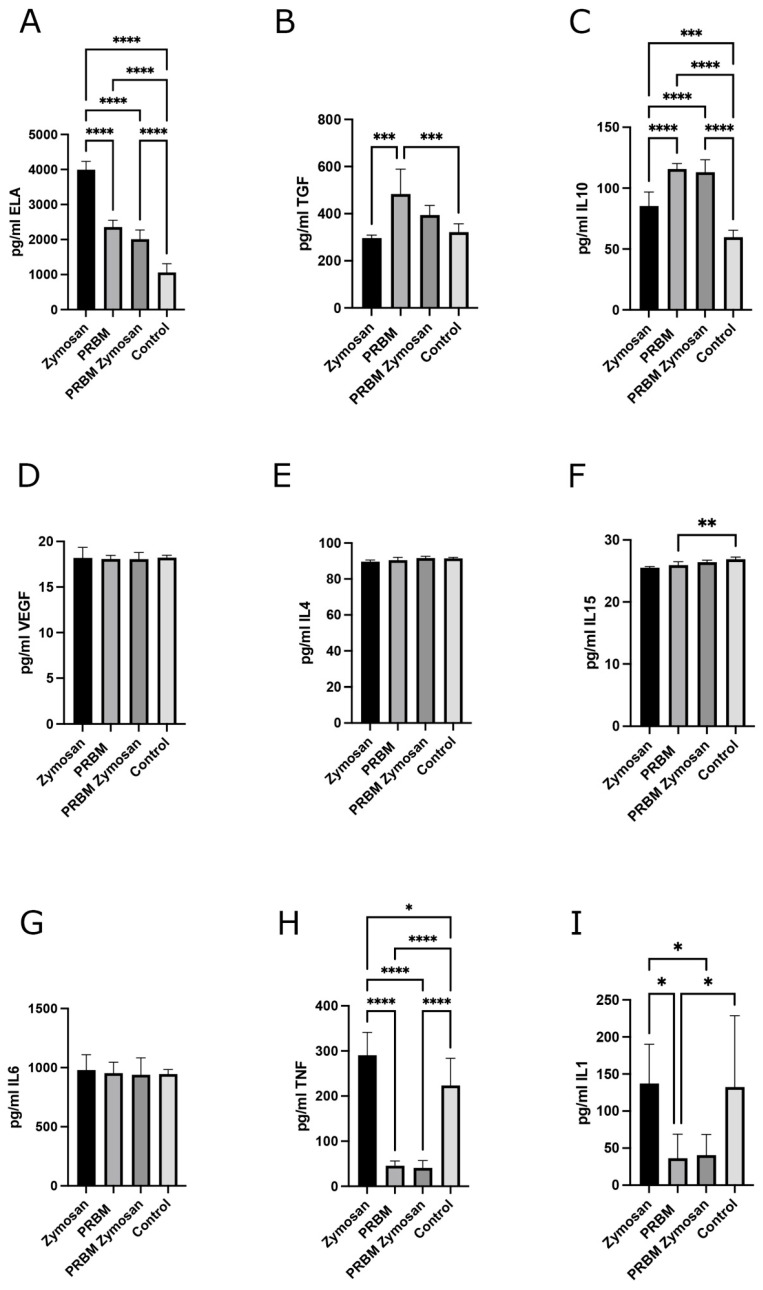
Evaluation of cytokines related to proinflammatory and regenerative phenotypes of PMNs after low-RCF PRF/PRBM combination for 4 h. The data show the absolute concentrations of ELA (**A**), TGFβ (**B**), IL10 (**C**), VEGF (**D**), IL4 (**E**), IL15 (**F**), IL6 (**G**), TNFα (**H**), and IL1 (**I**) in the appropriate experimental groups (PRF + Zymosan, PRF + PRBM, PRF + PRBM + Zymosan and low-RCF PRF control; six donors PRF, n = 6). * *p*-value < 0.05, ** *p*-value < 0.01, *** *p*-value < 0.001, and **** *p*-value < 0. 0001.

**Figure 4 biomedicines-13-01239-f004:**
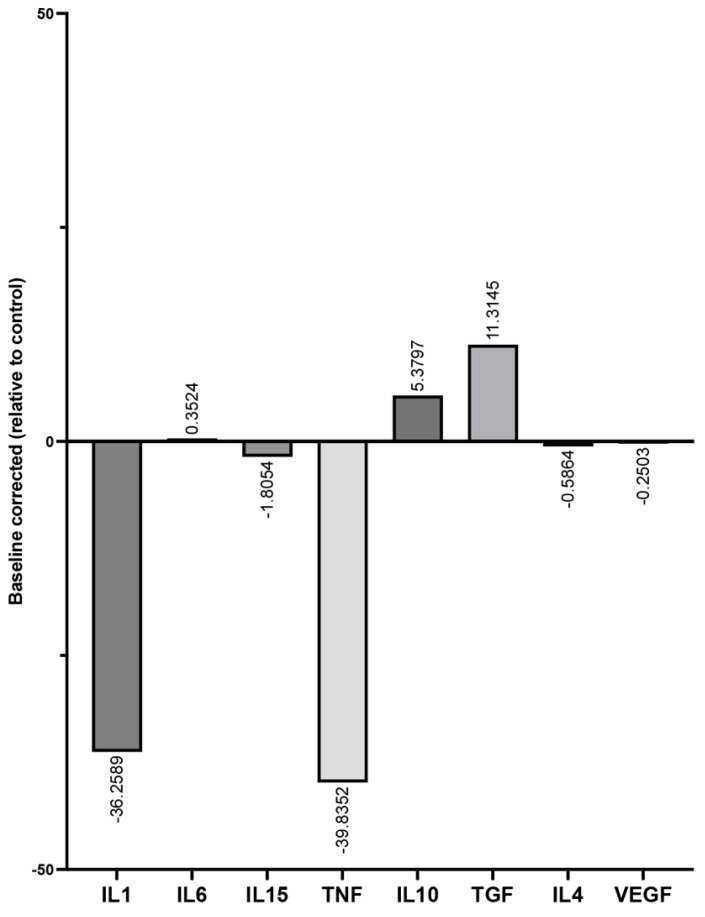
Summary of the PRBM-mediated cytokine profile (all analyzed cytokines) compared to the cytokine profile of control low-RCF PRF. The data are calculated relatively (as a percentage of the control; all tested donors; n = 6 and presented baseline corrected (control = baseline 0).

**Table 1 biomedicines-13-01239-t001:** Analyses of the recovery capacity of the different cytokines/growth factors. The percentage of cytokine recovery was calculated after 4 h of incubation (three different concentrations = input). GF = Cytokine/Growth factor.

CYTOKINES/ GROWTH FACTORS (GF)	[c] Input	4 h		
		[c] GF Measured	∆[c] GF	GF Recovery Rate [%]
**IL15**	62.5	24.38	−38.12	39.00%
	250	78.35	−171.65	31.34%
	1000	156.68	−843.32	15.67%
**VEGF**	125	20.96	−104.04	16.77%
	500	71.48	−428.52	14.30%
	2000	145.52	−1854.48	7.28%
**TGF**	125	0.51	−124.49	0.41%
	500	19.72	−480.28	3.94%
	2000	88.99	−1911.01	4.45%
**TNF**	62.5	27.61	−34.89	44.18%
	250	93.27	−156.73	37.31%
	1000	289.41	−710.59	28.94%
**IL1**	15.6	16.42	0.82	94.76%
	62.5	62.30	−0.20	99.68%
	250	256.36	6.36	97.45%
**IL4**	125	35.15	−89.85	28.12%
	500	123.37	−376.63	24.67%
	2000	861.11	−1138.89	43.06%
**IL6**	37.5	40.22	2.72	92.74%
	150	134.80	−15.20	89.87%
	600	576.51	−23.49	96.08%
**IL10**	125	37.30	−87.70	29.84%
	500	131.05	−368.95	26.21%
	2000	691.84	−1308.16	34.59%

## Data Availability

The data that support the findings of this study are available on request from the corresponding author (E.D.).

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
