# Peer review of "In Vitro Evaluation of the PMN Reaction on a Collagen-Based Purified Reconstituted Bilayer Matrix (PRBM) Using the Autologous Blood Concentrate PRF"

_biomedicines, 2025, doi:10.3390/biomedicines13051239_

Round 1
Reviewer 1 Report
Comments and Suggestions for Authors
Dear authors,
I have reviewed the manuscript and found the topic relevant and timely. The experimental design is overall appropriate, and the data are clearly presented. Below, I provide specific comments and suggestions aimed at improving the clarity and scientific robustness of your manuscript.
While the experimental design is overall solid and the data are clearly presented, the interpretation of cytokine modulation, particularly for TGFβ, IL10, and TNFα, would benefit from additional clarification. Specifically:
- Neutrophil Elastase (ELA) as a marker: The authors should briefly acknowledge that ELA is also present in degranulated or apoptotic neutrophils and does not exclusively indicate the presence of viable PMNs. Including an additional neutrophil marker (e.g., MPO or CD66b) or discussing this limitation would strengthen the characterization.
- Cytokine modulation vs. adsorption: In section 3.2, increases or decreases in cytokine levels (e.g., TGFβ and TNFα) are interpreted as evidence of PRBM-modulated PMN behaviour. However, in section 3.3, the cytokine recovery assays reveal substantial adsorption of these same cytokines by PRBM, with recovery rates below 10% (TGFβ) and 50% (TNFα, IL10). These findings suggest that the observed concentration changes might reflect physical sequestration rather than altered cellular secretion. The authors should revise or expand the discussion to address this possibility.
Clarifying these points will significantly enhance the robustness of the conclusions drawn from the cytokine data.
Additional Comment (Cellular Characterization): I suggest including complete blood count parameters from the original blood samples used for PRF preparation to further support the histological findings and cytokine release data. Specifically, quantification of red blood cells, platelets, and leukocyte subpopulations — as well as volume indices such as mean platelet volume and mean corpuscular volume — would provide valuable quantitative insights into the cellular composition of the different PRF preparations. This would enhance the interpretation of the observed differences between low and high RCF PRF and clarify whether cytokine variations are due to changes in cellular content or activation state.
Minor Comment: In Figure 2C, the unit of measurement for IL-6 appears to be incorrect or inconsistent with the other cytokines. Please double-check and correct the unit to ensure uniformity across the figure panels.
Author Response
Dear Reviewer,
we would like to thank you for carefully reading the manuscript and for your response. Please find included the resubmission of the manuscript with the revised title:
“In vitro evaluation of the PMN reaction on a collagen-based Purified Reconstituted Bilayer Matrix (PRBM) using the autologous blood concentrate PRF”
by Eva Dohle, Hongyu Zuo, BüÅŸra Bayrak, Anja Heselich, Birgit Schäfer, Robert Sader and Shahram Ghanaati
to be considered for publication as original research paper in Biomedicines. We have revised the manuscript according to your suggestions. The changes are highlighted in yellow color in the revised version of the manuscript and addressed in this letter. We would like to thank you for all your effort with the manuscript.
Yours sincerely,
Eva Dohle
General information:
The individual answers to the reviewer’s suggestions are addressed in this letter point by point. All changes in the revised manuscript have been highlighted in yellow colour.
Reviewer 1
Dear authors,
I have reviewed the manuscript and found the topic relevant and timely. The experimental design is overall appropriate, and the data are clearly presented. Below, I provide specific comments and suggestions aimed at improving the clarity and scientific robustness of your manuscript.
While the experimental design is overall solid and the data are clearly presented, the interpretation of cytokine modulation, particularly for TGFβ, IL10, and TNFα, would benefit from additional clarification. Specifically:
- Neutrophil Elastase (ELA) as a marker: The authors should briefly acknowledge that ELA is also present in degranulated or apoptotic neutrophils and does not exclusively indicate the presence of viable PMNs. Including an additional neutrophil marker (e.g., MPO or CD66b) or discussing this limitation would strengthen the characterization.
We would like to thank the reviewer for this suggestion. We would like to note that this information is already included in the results section of the manuscript:
„The PMN marker Neutrophil-Elastase (ELA; Fig. 1C/D), a serine protease stored in the azurophilic granules of neutrophils, is present in neutrophils at all stages of maturation, in circulating or degranulated PMNs and in neutrophil-derived extracellular traps (NETs).“According to this suggestion, we revised the discussion part of the manuscript and included this limitation.
- Cytokine modulation vs. adsorption: In section 3.2, increases or decreases in cytokine levels (e.g., TGFβ and TNFα) are interpreted as evidence of PRBM-modulated PMN behaviour. However, in section 3.3, the cytokine recovery assays reveal substantial adsorption of these same cytokines by PRBM, with recovery rates below 10% (TGFβ) and 50% (TNFα, IL10). These findings suggest that the observed concentration changes might reflect physical sequestration rather than altered cellular secretion. The authors should revise or expand the discussion to address this possibility.
We acknowledge the comment that the observed changes in cytokine concentrations may not solely reflect PRBM-induced modulation of PMN behavior, but could also be influenced by physical sequestration of cytokines by the PRBM matrix itself. In Section 3.3, our cytokine recovery assays revealed substantial adsorption effects, particularly for TGFβ (recovery rate <10%), TNFα, and IL-10 (both <50%). These findings suggest that the apparent decrease in pro-inflammatory cytokines such as TNFα, as well as the increase in anti-inflammatory or regenerative cytokines like TGFβ and IL-10, must be interpreted with caution. While we initially attributed these shifts to functional changes in PMN cytokine secretion profiles in response to PRBM, it is plausible that a portion of these effects result from the physical binding or retention of cytokines within the PRBM matrix, especially for TNF. This is especially relevant for cytokines with low recovery rates, where underestimation or overinterpretation may occur. Future studies employing alternative methods, such as immunohistochemistry, surface-bound cytokine quantification, or PRBM pre-saturation protocols, will be necessary to more precisely distinguish between true cellular modulation and adsorption-mediated depletion. We have revised the discussion accordingly to reflect these considerations and to provide a more nuanced interpretation of the observed cytokine patterns.
Clarifying these points will significantly enhance the robustness of the conclusions drawn from the cytokine data.
Additional Comment (Cellular Characterization): I suggest including complete blood count parameters from the original blood samples used for PRF preparation to further support the histological findings and cytokine release data. Specifically, quantification of red blood cells, platelets, and leukocyte subpopulations — as well as volume indices such as mean platelet volume and mean corpuscular volume — would provide valuable quantitative insights into the cellular composition of the different PRF preparations. This would enhance the interpretation of the observed differences between low and high RCF PRF and clarify whether cytokine variations are due to changes in cellular content or activation state.
We appreciate the reviewer’s thoughtful suggestion to include complete blood count parameters, including red blood cells, platelets, leukocyte subpopulations, and volume indices such as mean platelet volume and mean corpuscular volume, to further support our histological and cytokine findings. We fully agree that such data can provide valuable quantitative insight into the cellular composition and activation state of the different PRF preparations. However, these analyses have already been conducted and published in a number of previous work [reference], where we comprehensively characterized PRF fractions generated under varying relative centrifugal force (RCF) conditions. To maintain focus and avoid redundancy, we have not repeated those data here but have cited the relevant publication in the manuscript.
Minor Comment: In Figure 2C, the unit of measurement for IL-6 appears to be incorrect or inconsistent with the other cytokines. Please double-check and correct the unit to ensure uniformity across the figure panels.
This has been corrected in the revised version oft he manuscript.

Reviewer 2 Report
Comments and Suggestions for Authors
I would like to thank the author's for their efforts but I have some comments to be addressed
1. The introduction part should be improved and the back ground about collegen material and their application in various fields should be discussed.
2. The novelty of the work should be clearly expressed in the manuscript.
3. More references about collegen material application in medical Fields should be added. For example usage of collagen as nanofiber matrix membrane for wound healing application.
4. I would like to ask the author's to make ftir analysis to detect the components of collagen material.
Author Response
Dear Reviewer,
we would like to thank you for carefully reading the manuscript and for your response. Please find included the resubmission of the manuscript with the revised title:
“In vitro evaluation of the PMN reaction on a collagen-based Purified Reconstituted Bilayer Matrix (PRBM) using the autologous blood concentrate PRF”
by Eva Dohle, Hongyu Zuo, BüÅŸra Bayrak, Anja Heselich, Birgit Schäfer, Robert Sader and Shahram Ghanaati
to be considered for publication as original research paper in Biomedicines. We have revised the manuscript according to your suggestions. The changes are highlighted in yellow color in the revised version of the manuscript and addressed in this letter. We would like to thank you for all your effort with the manuscript.
Yours sincerely,
Eva Dohle
General information:
The individual answers to the reviewer’s suggestions are addressed in this letter point by point. All changes in the revised manuscript have been highlighted in yellow colour.
Reviewer 2
I would like to thank the author's for their efforts but I have some comments to be addressed
- The introduction part should be improved and the back ground about collegen material and their application in various fields should be discussed.
According to this suggestion, the manuscript has been revised with increased information on collagen materials and their application in various fields
- The novelty of the work should be clearly expressed in the manuscript.
The novelty of the work has been adressed more clearly in the revised version of the manuscript.
- More references about collegen material application in medical Fields should be added. For example usage of collagen as nanofiber matrix membrane for wound healing application.
Accordingly, we added more references regarding collagen material applications in the medical field including collagen-based nanofibres for wound healing applications.
(13) Dong, C.; Lv, Y. Application of Collagen Scaffold in Tissue Engineering: Recent Advances and New Perspectives. Polymers (Basel) 2016, 8 (2). DOI: 10.3390/polym 8020042 From NLM.
(15) Wang, Y.; Wang, Z.; Dong, Y. Collagen-Based Biomaterials for Tissue Engineering. ACS Biomaterials Science & Engineering 2023, 9 (3), 1132-1150. DOI: 10.1021/acsbiomaterials.2c00730.
(16) Mbese, Z.; Alven, S.; Aderibigbe, B. A. Collagen-Based Nanofibers for Skin Regeneration and Wound Dressing Applications. Polymers (Basel) 2021, 13 (24). DOI: 10.3390/polym13244368 From NLM.
(17) Law, J. X.; Liau, L. L.; Saim, A.; Yang, Y.; Idrus, R. Electrospun Collagen Nanofibers and Their Applications in Skin Tissue Engineering. Tissue Eng Regen Med 2017, 14 (6), 699-718. DOI: 10.1007/s13770-017-0075-9 From NLM.
- I would like to ask the author's to make ftir analysis to detect the components of collagen material.
We would like to thank the reviewer for this suggestion. The used collagen-based Purified Reconstituted Bilayer Matrix (PRBM) Geistlich Mucograft® is a certified, medical-grade product made from porcine collagen (primarily Type I and III). Its composition is already well-documented and standardized by the manufacturer (Geistlich Pharma). Therefore, repeating FTIR for every application or paper might be seen as redundant.

Round 2
Reviewer 1 Report
Comments and Suggestions for Authors
Thank you for the authors' responses and the revised manuscript. While not all of my suggestions were fully implemented, the authors have acknowledged the main points and addressed them appropriately as limitations of the study. I am satisfied with the revisions made.
Reviewer 2 Report
Comments and Suggestions for Authors
I would like to thank the author's for their efforts and the revised copy is ok